# Faecal Microbiota Transplantation and Chronic Kidney Disease

**DOI:** 10.3390/nu14122528

**Published:** 2022-06-17

**Authors:** Ji Bian, Ann Liebert, Brian Bicknell, Xin-Ming Chen, Chunling Huang, Carol A. Pollock

**Affiliations:** 1Kolling Institute, Sydney Medical School, Faculty of Medicine and Health, University of Sydney, Royal North Shore Hospital, St Leonards, NSW 2065, Australia; jbia3972@uni.sydney.edu.au (J.B.); xin-ming.chen@sydney.edu.au (X.-M.C.); 2Faculty of Medicine and Health, University of Sydney, Camperdown, NSW 2006, Australia; ann@pbmconsults.com; 3College of Health and Medicine, Australian National University, Deacon, ACT 2600, Australia; brian@pbmconsults.com

**Keywords:** faecal microbiota transplantation, chronic kidney disease, gut microbial metabolites, gut barrier, immunity/inflammation

## Abstract

Faecal microbiota transplantation (FMT) has attracted increasing attention as an intervention in many clinical conditions, including autoimmune, enteroendocrine, gastroenterological, and neurological diseases. For years, FMT has been an effective second-line treatment for *Clostridium difficile* infection (CDI) with beneficial outcomes. FMT is also promising in improving bowel diseases, such as ulcerative colitis (UC). Pre-clinical and clinical studies suggest that this microbiota-based intervention may influence the development and progression of chronic kidney disease (CKD) via modifying a dysregulated gut–kidney axis. Despite the high morbidity and mortality due to CKD, there are limited options for treatment until end-stage kidney disease occurs, which results in death, dialysis, or kidney transplantation. This imposes a significant financial and health burden on the individual, their families and careers, and the health system. Recent studies have suggested that strategies to reverse gut dysbiosis using FMT are a promising therapy in CKD. This review summarises the preclinical and clinical evidence and postulates the potential therapeutic effect of FMT in the management of CKD.

## 1. Introduction

Chronic kidney disease (CKD) affects approximately 10% of adults worldwide [1]. It has been identified as a significant health issue, which has been driven by the epidemic trend of diabetes, hypertension, and the dynamics of the population aging process [2]. CKD is a continuum of chronic changes in the kidney, including chronic inflammation, tubulointerstitial fibrosis, glomerulosclerosis, and vascular rarefaction. Intensive research is ongoing to unravel the pathogenesis of CKD, which should provide more opportunity to slow CKD progression by non-pharmacologic and novel pharmacologic interventions. Blockade of the renin–angiotensin–aldosterone systems has been shown to slow the progression of proteinuric kidney disease [3,4]. More recently, type 2 sodium–glucose linked transporter (SGLT2) inhibitors in cardiovascular and renal specific outcomes trials have been shown to further limit CKD progression to end-stage kidney disease in diabetic and, in the DAPA-CKD trial, non-diabetic patients [5,6]. The only effective treatment for end-stage kidney disease is kidney replacement therapy, including dialysis and kidney transplantation. As CKD advances, associated mortality, loss of life quality, and increased hospitalization lead to a dramatically ascending financial burden on the public healthcare system, communities, patients, and families [7]. The stage and progression of CKD may change the absorption, distribution, metabolism/transport, and excretion of medications [8,9,10]. Hence, novel treatments have been urgently needed to prevent and treat CKD. Over the past decade, mounting studies have identified the role of the gut microbiota in a wide range of inflammation-associated clinical conditions, including *Clostridium difficile* infection (CDI) induced diarrhea [11], irritable bowel disease (IBD) [12], obesity, and diabetes [9]. These studies have led to the hypothesis that targeting the gut microbiota may be a potential strategy to manage CKD, which is increasingly regarded as an inflammatory disease [13].

The gut microbiota is the broad spectrum of microorganisms accommodated in the gastrointestinal (GI) tract, consisting of bacteria, viruses, archaea, protists, and fungi. A healthy human GI tract harbors more than 10^14^ bacteria, which encode 100 times more genetic information than the human genome [14]. With the availability of cheaper and faster biotechnologies, such as high-throughput next-generation sequencing, features of the gut microbiome, including composition, abundance, diversity, and functions, have been explored [15]. The current understanding of the gut microbiome community is primarily based on the taxonomic characteristics at genus levels due to relatively lower sequencing costs. Applying the bacterial 16S ribosomal RNA (16S rRNA) gene sequencing is a successful strategy for genus identification and characterization, which is independent of labour-intensive culture-based techniques [14]. Alternatively, gut microbiota can be analysed at species levels via metataxonomics, which is a deeper sequencing strategy and has more power to discover those bacteria with less abundance [16]. In healthy adults, most of the gut ecosystem is primarily dominated by *Firmicutes* and *Bacteroides*, with a relatively low population of *Proteobacteria*, *Actinobacteria*, *Fusobacteria*, *Verrucomicrobia*, etc. [17]. In addition, the homeostatic gut microbiome community serves the host via a wide range of physiological activities, including digesting food [18,19], preventing pathogens [20], maintaining the function and integrity of the intestinal epithelial layer [21,22], and modulating the host immune system [23,24,25]. With the assistance of advanced bioinformatics and machine learning artificial intelligence, strong links have been identified between the gut microbiota and many human diseases [26], such as diabetes [27], irritable bowel disease [28], and autism [29]. Increasing studies have identified microbiota-derived metabolites as key molecular mediators between the microbiota and host, mainly short-chain fatty acids (SCFAs) [30], bile acids [31], branched-chain amino acids [32], trimethylamine N-oxide [33], tryptophan, and indole derivatives [34]. 

Appreciation of the importance of the gut microbiome for human health and disease derives from clinical practices affecting gut bacteria, such as antibiotic use. Disturbance of healthy gut microbiota, also termed gut dysbiosis, contributes to the pathogenesis of many clinical diseases, such as metabolic disorders [35], inflammatory bowel disease [36], neurodegenerative diseases [37], and CKD [38,39]. Recently, gut dysbiosis has been suggested to play a pathogenic role in CKD via modulation of the immune system, impairment of gut barrier integrity, and gut microbiota-derived metabolites [40]. Gut bacteria-derived metabolites, such as short-chain fatty acids (SCFAs), uremic toxins, endotoxins, etc., have also shown a close association with different severities of CKD, indicating a potential important role of gut metabolites in the progression of CKD [41]. 

As a potent microbiome-based invention, faecal microbiota transplantation (FMT) transfers the gut microbial community from healthy donors to recipients to restore gut microbiota homeostasis. FMT is an established therapy for recurrent CDI, with a treatment efficacy above 85% [42]. Moreover, accumulating studies have shown the benefit of FMT for several diseases associated with dysregulated gut microbiota, including autism, cancer, and type 2 diabetes [7,43,44,45,46]. Although studies using FMT to prevent or treat CKD are still limited, preliminary evidence of the efficacy of FMT indicates that the application of FMT may offer a promising alternative for the treatment of CKD.

This review will demonstrate gut dysbiosis occurring in CKD and summarise its predominant pathogenic mechanisms that contribute to progressive kidney dysfunction, including the regulation of the immune system, gut microbiota-derived metabolites, renin–angiotensin system, and gut barrier. The application of FMT in CKD as a potential therapeutic option will also be discussed.

## 2. Evidence of Gut Microbiota Dysbiosis in CKD

Gut dysbiosis is both a quantitative and qualitative alteration in the composition and metabolic profiles of the gut microbiome. Increasing evidence has confirmed gut dysbiosis occurs at the onset of CKD, including in the development of CKD and progression to end-stage kidney disease (ESKD) [47,48,49]. Recent pre-clinical and clinical studies exploring gut microbiota features in CKD are summarized in Table 1, focusing on the relative abundance of primary bacterial groups, such as *Firmicutes, Bacteroidetes, Actinobacteria*, *Proteobacteria*, *Fusobacteria*, and *Verrucomicrobia* compared to controls. 

### 2.1. Animal Studies

In the diabetic kidney disease (DKD) mouse model, the microbiome analysis of operational taxonomic units (OTUs) identified an increased abundance of *Firmicutes, Proteobacteria*, and *Verrucomicrobia* but a significantly lower presence of *Bacteriodetes* in the DKD group compared to the healthy controls [56]. Similarly, in an adenine-induced CKD rat model, there was an enrichment of *Firmicutes* but a lower abundance of *Bacteroides* in the CKD group compared to the normal rats. Additionally, a reduced amount of *Ruminococcaceae UCG-014*, *Prevotellaceae*_*NK3B31_group*, *Ruminococcus* 1, *Lachnospiraceae UCG-001*, and *Clostridium leptum* group bacteria have been found in the adenine-induced CKD rat or mouse models, respectively [52]. 

### 2.2. Human Studies 

In a recent systematic review incorporating twenty-five studies with 1436 CKD patients and 918 healthy adults, Zhao et al. found that the gut microbiome in CKD patients had an increased abundance of phylums *Fusobacteria* and *Proteobacteria*, genera *Streptococcus*, *Desulfovibrio*, and *Escherichia_Shigella*, and a reduced presence of genera *Faecalibacterium*, *Roseburia*, *Pyramidobacter*, *Prevotella_*9, and *Prevotellaceae_UCG-001*. Moreover, elevated levels of bacterial metabolites trimethylamine n-oxide (TMAO) and p-cresol sulfate (PCS) and lower production of SCFAs were found in CKD patients [55]. A study on 29 Chinese patients with immunoglobulin A nephropathy (IgAN) also showed a reduced amount of *Clostridia*, *Eubacterium*, and *Alistipes* compared with healthy controls, indicating the pathogenic role of depletion of SCFA-producing bacteria in systemic inflammation and IgAN progression [53]. In patients with IgAN, a higher abundance of class *Coriobacteriia* and genera *Legionella*, *Enhydrobacter*, and *Parabacteroides* was present in the blood, and genera *Bacteroides*, *Escherichia-Shigella*, and *Ruminococcus* in stool in comparison with healthy controls [51]. Additionally, a clinical study in ESKD patients identified six taxa, including *Akkermansia*, *Dialester*, *Enterococcus*, *Rominicoccus*, *Blautia*, and *Bacteroides*, that contributed to the elevated levels of uraemic toxins [57]. Patients with DKD had a higher abundance of *Verrucomicrobia*, *Proteobacteria*, and a lower amount of *Bacteroidetes* at the phylum level, with *Escherichia-Shigella*, *Enter Klebsiella*, *Streptococcus*, *Akkermansia*, and *lactobacillus* increased at the genus level. In particular, SCFA-producing bacteria, such as *Blautia* and *Faecalibacterium*, were decreased in DKD patients compared to healthy controls [54]. Kim et al. analysed the gut microbiome from 103 patients with CKD (stage 1 to 5) and 46 healthy controls. The results showed an increased abundance of *Oscillibacter* and *Alistipes* and a decreased abundance of *Veillonella*, *Lachnospira*, and *Dialister* in CKD patients compared to the healthy controls [58]. Another clinical study that enrolled 14 healthy controls and 138 CKD patients indicated a lower proportion of SCFA-producing bacteria, including *Bifidobacterium* and *Streptococcus*, in CKD patients. Interestingly, kidney function loss was positively related to a higher proportion of *Enterobacteriaceae* and *E. coli* [50].

## 3. The Role of the Gut Microbiota in CKD

In CKD, the gut–kidney axis is potently affected by the gut microbiota in a bidirectional manner [59]. CKD increases the retention of metabolic waste products, gut bacteria-derived metabolites (i.e., uremic toxins) with concomitant medications, poor nutrition, and comorbidities contributing to the altered gut microbiota. Conversely, gut dysbiosis contributes to progressive CKD mainly via maladaptive immune responses, microbial metabolites that are harmful to the kidneys, activation of the RAS, and a disruption of the gut barrier, allowing gut toxins and metabolites to enter the systemic circulation. Thus, a vicious cyclical interaction occurs between the gut and kidney, amplifying both organs’ injury. 

### 3.1. Immunomodulatory Mechanisms of Gut Microbiota in CKD

It is postulated that gut dysbiosis promotes CKD by modulating the host’s immunity, which depends on the activation of the host’s innate and adaptive immune responses [60]. Previous studies have shown that host defence systems, such as innate and adaptive immunity, could coevolve, be trained, and tolerated reciprocally with the prevailing gut microbiota [61]. This intricate and symbiotic interaction between microbiota and host immune systems significantly affects the health status of animals and humans. 

Gut microbiota dysbiosis contributes to the impaired gut barrier, excessive production of uremic toxins, and translocation of the gut microbiome, leading to the abnormal activation of the immune cells [62]. The active immune cells, such as neutrophils, macrophages, dendritic cells, and lymphocytes, then infiltrate kidneys and promote the pro- or anti-inflammatory reactions [60]. Wang et al. found that neutrophils and macrophages were recruited and activated as part of the first-line immune response in innate immunity against kidney disease induced by adenine in mice [63]. Moreover, Funken et al. identified that the γδ-T cells as potent mediators in ischemia-reperfusion injury (IRI) contributed to the increased influx of neutrophils and pro-inflammatory cytokines, which were not found in δ-T cell deficiency mice [64]. IRI mouse model induced by bilateral renal pedicle clamping resulted in altered gut bacterial dysbiosis resulting in significant decreases in *Lactobacilli*, *Ruminococacceae*, and an increase in *Enterobacteriaceae*. Depleting gut bacteria by oral antibiotics or FMT derived from sham-operated mice prevented mice from developing IRI, which was confirmed to be associated with reduced Th 17, Th 1 activation, and the expansion of M2 macrophage and Treg [65]. Germ-free animal models have shown a significant increase in the population of T helper 17 (T_H_17) cells but a deficit in the immune-mediating regulatory T (Treg) cells, leading to susceptibility to colitis after FMT from IBD patients [66]. Notably, a bacterial polysaccharide from segmented filamentous *Bacteroides fragilis* could ‘educate’ the host immune system to be tolerant to residing microbes, primarily by modulating CD4+ Tregs to secrete IL-10, which was dependent on Toll-like receptor (TLR) in mice [67]. T_H_17 cells are present in high density in the gut, and their presence and activation rely on the adjacent gut microbiota. Krebs and his team directly identified that microbiota-induced T_H_17 cells migrated from the gut and accumulated in kidneys, which was mediated by the chemokine C-C motif chemokine ligand 20 (CCL20) and its receptor C-C Motif Chemokine Receptor 6 (CCR6) in autoimmune kidney disease [68]. 

Individual compartmentalized gut-associated lymphoid tissues (GALTs) inhabit the interface between the intestinal lymphatic and blood systems, allowing mature immune cells constant access to the epithelial layers and lamina propria. In humans, the GALT interplay with the gut microbiota to exert immune responses and tolerance [69]. However, a recent study strongly supported that GALT is a significant induction site of IgAN [70].

### 3.2. Gut Microbial Metabolites and CKD

Due to the relative richness of proteolytic bacteria, gut bacterial metabolites are profoundly altered in CKD, resulting in reduced local and systemic SCFAs and increased uremic toxins [71]. 

SCFAs are aliphatic carboxylic acids with fewer than six carbon (c) atoms produced by bacteria inhabiting the colon after fermentation of dietary fibre or protein catabolism. In humans, the major SCFAs are acetate, propionate, and butyrate, which are produced by prebiotic fermentation via specific gut bacteria, such as *Blautia, Bacteroides, Butyricoccus, Bifidobacterium, Prevotella, Megasphaera*, and *Butyrivibrio* [72] in the intestine and colon [73]. In previous studies, metabolites derived from the gut microbiota via fermentation of dietary fibre have been shown to directly influence the composition and function of the host immune system [74,75,76]. SCFAs activate G-protein-coupled receptors (GPR), including GPR43, GPR109A, GPR41, and O1f78, etc., thereby attenuating reactive oxygen species (ROS), inflammation [77], and mitigating fibrosis in the kidney [48]. SCFAs preferably activate GPRs; for example, acetate activates GPR43, propionate activates GPR41, while GPR43 and butyrate activate GPR41 [78].

In a cohort clinical study, Wang et al. measured the levels of SCFAs in 127 patients with CKD and 63 healthy controls. The results showed that the serum SCFAs levels, especially butyrate, were significantly decreased in CKD patients compared with healthy controls, and supplementation of butyrate might slow the progression of CKD [79]. An investigation that enrolled 105 children and adolescents with CKD found a difference in plasma acetate between children with and without hypertension, suggesting its preventive role for hypertension in children with CKD. In addition, reduced plasma butyrate was found in the advanced and stable blood pressure groups at the 1-year follow up, which may potentially be caused by the decline in eGFR [80]. In addition, Chai et al. identified substantially lower production of SCFAs, including acetic acid, propionic acid, butyric acid, and iso-butyric acid, in IgAN patients compared with those in the control group [53]. Furthermore, a recent clinical study recruiting 30 patients with DKD confirmed that SCFAs were reduced in serum and stool compared to 30 normal controls [81]. In line with those clinical findings, the decreased plasma levels of SCFAs and impaired renal function were observed in a rat model of CKD induced by dietary adenine, which was attenuated by gum acacia [82]. In a DKD mouse model induced by streptozotocin (STZ), Li et al. reported that a high-fibre diet significantly reduced albuminuria, kidney hypertrophy, glomerular injury, tubulointerstitial fibrosis, and interstitial macrophage infiltration compared with healthy controls. They further confirmed that those benefits were associated with improved gut dysbiosis, increased SCFA-producing bacteria, and boosted SCFAs production. Supplementation with SCFA further confirmed that SCFA is responsible for the renoprotective effect of dietary fibre, which was mediated through the activation of GPR43 or GPR109A [83]. 

The accumulation of serum uremic toxins due to gut microbiota dysregulation is strongly associated with oxidative stress and systemic inflammation, which is increasingly recognized to play a pivotal role in the onset and progression of CKD [84]. Protein-bound uremic toxins (PBUTs) are produced by gut bacteria by the metabolism of aromatic amino acids and are not efficiently removed by dialysis. The most investigated PBUTs derived from the gut microbiota are indoxyl sulfate (IS) and para-cresyl sulfate (pCS). IS is synthesized from dietary tryptophan, and pCS is metabolized from dietary phenylalanine and tyrosine via gut bacteria. IS and pCS were negatively correlated to the estimated glomerular filtration rate through all stages of CKD among children and adolescents [85]. Similarly, another clinical study recruiting 342 patients with CKD measured total and free serum IS and pCS by ultra-performance liquid chromatography-tandem mass spectrometry (UPLC-MS/MS) and found a positive correlation between the analyte and CKD progressions [86]. Both PBUTs can trigger and exacerbate tubulointerstitial fibrosis and glomerular sclerosis, impairing kidney function in vivo [87,88]. 

In addition to IS and pCS, trimethylamine-N-oxide (TMAO) has been recognized as a gut microbiota-derived uremic retention product associated with higher mortality and risk of cardiovascular events [33,89]. TMAO is produced from the metabolism of quaternary amines such as betaine, L-carnitine, and phosphatidylcholine, which are precursors of trimethylamine [90]. Food sources of choline and carnitine, such as cheese and red meat, are metabolized into trimethylamine mainly by gut microorganisms such as *Firmicutes* and *Proteobacteria* [91]. Elevated serum TMAO was identified as positively related to mortality in 521 patients with CKD studies over a 5-year follow-up [89]. Furthermore, an observational prospective cohort study of peritoneal dialysis (PD) patients (*n* = 105) demonstrated that elevated plasma TMAO induces peritoneal inflammation and increases the risk of peritonitis in patients undergoing PD [92]. 

### 3.3. Renin–Angiotensin System (RAS) and CKD

The renin–angiotensin system (RAS) has been recently confirmed as a critical link between gut microbiota and CKD [93]. RAS inhibitors are prescribed to patients to decrease proteinuria and defer CKD progression [94]. However, uremic toxins, including IS, pCS, and TMAO, can activate the RAS [95,96], leading to increased intraglomerular pressure and excessive local release of pro-inflammatory factors [97]. Sun et al. reported that IS and pCS could induce kidney fibrosis via activating the RAS/pro-inflammatory transforming growth factor-β1 (TGF-β1) axis in half-nephrectomized-induced mouse CKD models [95]. Similarly, Jiang et al. found that TMAO augmented angiotensin II-induced vasoconstriction and promoted angiotensin II-induced hypertension through the PERK/ROS/CaMKII/PLCβ3 axis [96]. Consistent with this, another recent study in the rat DKD model confirmed that gut microbiota disorders contributed to kidney injuries via activating the RAS, as evidenced by significantly increased expressions of angiotensin II, angiotensin-converting enzyme and the angiotensin II type 1 receptor in DKD groups compared to the healthy controls [98]. Accumulating studies have explored the potential therapeutic strategy for treating CKD by suppressing the gut dysbiosis-induced activation of the RAS. For example, Chen et al. demonstrated that Alisol B 23-acetate, isolated from Alisma Orientale, can attenuate CKD progression via modulating the RAS and gut–kidney axis in 5/6 nephrectomized and unilateral ureteral obstructed rat models [99].

### 3.4. Disrupted Gut Barrier and CKD

Healthy gut microbiota homeostasis may assist with the development of host immunity, while a disturbed gut microbiome hampers and damages it [100]. Gut dysbiosis may manipulate host immunity/inflammation directly through its interplay with the adjacent epithelial barrier of the gastrointestinal (GI) tract [101] and the mucus layer secreted by mucosal epithelia, the mucosa, and resident immune cells [102]. Vaziri et al. identified the critical role of the intestinal barrier in a rat model of 5/6 nephrectomy-induced CKD. They demonstrated that uremia impaired the integrity of the intestinal barrier, as evidenced by the reduction in the key proteins expressed in the colonic mucosa, including claudins-1, occludin, and zonula occludens-1 (ZO-1). They postulated that the impaired intestinal barrier function due to the disintegration of the colonic tight junction might be a key contributor to the systemic inflammation in CKD [103]. Similarly, Yang et al. confirmed that gut microbiota dysbiosis in CKD was positively associated with the severity of the impaired gut barrier and abnormal intestinal immunity in the mucosal layer in 5/6 nephrectomized mice [104]. Decreased expression of tight junction proteins, gradual loss of colonocytes, and gut dysbiosis were also positively correlated with the impaired gut barrier in the mouse model of IRI [65]. Ji et al. reported that the treatment with rhubarb (a traditional Chinese herbal medicine) improved gut microbiota dysbiosis, suppressed systemic inflammation, and attenuated kidney fibrosis in 5/6 nephrectomized rats through restoring the intestinal barrier. These positive outcomes suggest that the intestinal barrier is an essential target for CKD treatment [105].

CKD can further impair gut barrier function due to decreased SCFAs production and the accumulation of uremic toxins, which contributes to inflammation, oxidative stress and further damages the gut epithelial monolayer. A reduction in SCFAs and excessive production of uremic toxins lead to a downward spiral of worsening CKD in association with worsening gut function [87,106]. CKD then promotes malnutrition, which further impairs the gut epithelial layer due to fluid retention, metabolic acidosis, and uric acid accumulation [107]. Microbiota depletion with multiple broad-spectrum antibiotics, including ampicillin, neomycin sulfate, metronidazole, and vancomycin, often used in patients with CKD, can further impair gut barrier function [108]. Conversely, supplementation with the SCFA sodium butyrate to db/db mice significantly increased gut epithelial integrity with increased expression of intercellular adhesion molecules, such as ZO-1 [109].

## 4. Faecal Microbiota Transplantation (FMT)

FMT potently ‘rebuilds’ the gut microflora’s dynamic homeostasis, structure, and diversity by transferring healthy gut microbiota into recipients. The significant relationship between gut microbiota dysregulation and many clinical conditions has enabled FMT to treat multiple clinical conditions, such as autoimmune [110], neurodegenerative [111], CDI [112], and metabolic disorders [113,114,115]. The most successful clinical application of FMT is currently to treat CDI with up to a 90% success rate and with a good safety profile [116], as no short- or long-term adverse effects have been described [117]. In addition, FMT can be delivered to the recipients’ GI tract in several ways, including capsules, naso-enteric tube, and colonoscopy [118].

FMT plays a vital role in shaping and modulating the immune system, significantly impacting the peripheral immune system and inflammatory responses. Burrello et al. demonstrated that FMT could attenuate inflammatory responses in the colon and restore gut homeostasis via activating different immunoregulatory signalling pathways. Consequently, FMT increased IL-10 secretion from innate and adaptive immune cells and inhibited the activity of monocytes, dendritic cells, and macrophages to acquire, process, and present antigens to T cells inhabiting the intestine [119]. In another colitis mouse model induced by dextran sodium sulfate (DSS), the abundance of colonic CD4+, CD8+, and invariant natural killer T (iNKT) cells were significantly reduced in the FMT group compared to the non-FMT and control group. In addition, colonic expression levels of crucial cytokine genes *Il6*, *Il7*, *Ifng*, *Tnf*, and *Il1b* were downregulated significantly compared with the non-FMT group [120].

FMT has been used as a “brute-force” therapeutic strategy to restore or improve gut homeostasis mediated through the regulation of gut bacteria-derived metabolites. In mouse models with atopic dermatitis (AD), gut microbiota dysbiosis was restored to the donor state together with elevated concentrations of SCFAs [121]. FMT also corrected the gut bacterial imbalance in traumatic brain injury rat models and reduced gut bacteria-synthesized uremic toxins, such as TMAO [122]. In a traumatic brain injury rat model, FMT could reduce the concentration of TMAO in the ipsilateral brain and serum. In contrast, the TMAO level was lowered in the faeces via restoring gut microbiota dysbiosis [122]. In addition, FMT strongly reduced systemic endotoxemia in a sepsis rat model, leading to beneficial effects on improving gut barrier functions [123].

Although a wide variety of the gut microbiota-based approaches, such as probiotics and prebiotics, have shown positive outcomes in managing CKD, a considerable number of studies generated controversial results [124,125,126]. Hence, FMT, which restores the overall gut microbiome community, might offer a promising alternative to protect kidneys from injury. It is worth exploring mechanisms accounting for those benefits from FMT in various clinical and pre-clinical contexts.

## 5. FMT Studies in CKD

Accumulating evidence has highlighted the pathogenic role of gut microbiota dysbiosis in the setting and progression of CKD [127,128,129], which has led researchers to investigate animal studies of FMT in CKD further, either alone or as adjunctive therapy (Table 2).

FMT from patients, donors or CKD rodent models has been used to investigate the correlation between gut dysbiosis and CKD progression. For example, Wang et al. administered FMT from patients with ESKD or healthy donors to germ-free CKD mice or antibiotic-treated CKD rats. They found that the gut microbiome in the recipient mice was modulated and exhibited the features of the ESKD patients or control donors. The gut microbiota from patients with ESKD increased the production of uraemic toxins, aggravated interstitial fibrosis, and oxidative stress, which further confirmed the causative relationship between the aberrant gut microbiota and kidney disease development [130]. Consistently, similar findings were observed in FMT experiments in STZ-induced DKD mice. Higher levels of TMAO and lipopolysaccharide (LPS), different microbiota constituents, and more significant kidney damage were found in the group receiving FMT from mice with severe proteinuria compared to those receiving FMT from mice with mild proteinuria, indicating that the differences in the gut microbiome may contribute to kidney disease severity [131].

In addition, FMT has been a helpful tool to demonstrate that the modulation of the intestinal microbiota may account for one of the key mechanisms mediating the renal protective effect of therapeutic interventions for CKD, as demonstrated in various FMT studies. By transplanting the resveratrol-modified faecal microbial, Cai et al. confirmed that resveratrol, a stilbene polyphenolic compound, exerted its renal protective effect by improving the faecal microbial community. They reported that FMT from healthy resveratrol-treated control mice restored the gut microbiome and regulated intestinal permeability accompanied by an inhibited inflammatory response and improved renal function [132]. Similarly, in the mechanistic study for *Astragalus membranaceus* and *Salvia miltiorrhiza* (AS), the administration of FMT from the *AS*-treated group attenuated cyclosporin A-induced kidney damage and fatty acid metabolism through modulation of the “gut–kidney axis.” Those benefits were positively correlated with an improved intestinal barrier, restored intestinal flora structure, increased abundance of bacteria producing butyric acid and lactic acid, and improvement in the miRNA–mRNA interaction profiles related to Butanoate and Tryptophan metabolism [133].

FMT, using a homeostatic gut microbiome collected from healthy donors, has been investigated as a potential therapeutic strategy for managing CKD. In a mouse model of adenine-induced CKD, FMT significantly reduced the production of uremic toxins derived from the intestinal cresol pathway with a profound improvement of gut microbiota disturbance, as evidenced by the increased alpha diversity. These studies suggested that FMT is an effective intervention for gut microbiota [134]. Hu et al. recently applied FMT from the healthy rats into rats induced to diabetes with STZ. The results showed that transplantation of microbiota from healthy rats potently ameliorated tubulointerstitial injury in the diabetic group associated with restoring dysregulated cholesterol homeostasis with decreased serum triglycerides and less lipid accumulation in the kidneys of diabetic rats, which was mediated through the activation of GPR43 [135]. Recently, Lu et al. reported that antibiotic treatment mediated gut microbiota depletion and treatment with FMT from healthy control rats effectively increased podocyte insulin sensitivity and alleviated glomerular injury in diabetic rats. This was associated with downregulation of GPR43 expression, suggesting an essential role of gut microbiota-modulated GPR43 in STZ-induced diabetic kidney disease [136]. Most recently, Bastos, R.M. et al. further confirmed the safety and efficacy of FMT as an effective non-pharmacological approach in BTBR^ob/ob^ DKD mice [137].

Interestingly, instead of transplantation of the whole gut microbiome, Zheng et al. constructed a bacterial micro-ecosystem (BME) with a small bacterial consortium encapsulated in calcium alginate microspheres coated with a layer of polydopamine nanofilm. The administration of a microencapsulated bacterial cocktail consisting of three different bacteria, including *Escherichia.coli, Bacillus subtilis*, and *Lactobacllus acidophilus*, showed benefits without any evident adverse effects by eliminating nitrogenous waste products in multiple animal models of kidney failure, including murine AKI, murine chronic kidney failure, and porcine kidney failure models. Importantly, this study provides a direction for microbiome-based therapies using a bioengineering strategy [138].

FMT has also been applied in clinical case studies to improve kidney function (Table 2). In a clinical case report, FMT extracted from a healthy male eligible donor was administered endoscopically to treat a patient with membranous nephropathy. After two treatments, FMT ameliorated the associated nephrotic syndrome and improved kidney function with increased total serum protein and albumin levels as well as decreased serum creatinine and 24 h urine protein [139]. Two patients with IgAN received FMT regularly via transendoscopic enteral tubing for 6–7 months. FMT treatment decreased 24 h urinary protein, increased serum albumin, and restored gut microbiota in both patients [140]. Given that FMT could potently alter gut microbiota’s overall composition and function, as reviewed above, FMT may benefit CKD via several classic mechanisms (Figure 1).

## 6. Conclusions and Future Studies

This review summarises the role of the gut microbiome in the pathogenesis of CKD, the primary mechanisms of FMT as a novel microbiota-based intervention and, most importantly, an update in animal and clinical studies on managing CKD using FMT. The predominant effects of FMT on CKD include modulating immune responses, gut microbiota metabolites, the RAS axis, and gut barrier function. Although FMT has yielded clinical benefits in several diseases, its utilization in extended clinical practice may be limited due to numerous factors, such as the definition of the healthy donor, screening procedures, sample preparation, storage condition, dose–response, methods of delivery, settings, and progression of different diseases. Clinical studies using FMT in patients with CKD are primarily small-scale and underpowered, as they were generally designed as pilot studies for safety reasons.

It is well recognized that “one stool does not fit all”. Hence, donor selection and administration methods in specific diseases are dominant determinants of FMT success rates. Therefore, decoding FMT strategies have to rely on the advantage of artificial intelligence, advanced bioinformatic techniques, and machine learning algorithms, such as multi-omics correlation analysis. A set of parameters for the gut microbiota, plasma, uraemic toxins, faecal metabolites, and kidney function need to be linked and systemically analysed, which will strongly support researchers in understanding the role of FMT and standardizing procedures and protocol evolution in the future.

Considering the potential pathogenic factors derived from overall gut microbiome transplantation, novel FMT consisting of several specific bacterial strains extracted from healthy stool might be an alternative intervention with much lower risks. For example, FMT could be further optimized in alternative ways, such as bacterial consortia and BME, to reduce potential pathogenic factors before application in patients to improve or slow CKD progression. Bacterial consortia are derived from culturing selected gut bacteria from the donor faecal samples, allowing for mass production, increased consistency, and thorough bacterial community characterization, including eliminating pathogenic microorganisms. Biomedical engineering techniques could also contribute to FMT development.

To conclude, standardization of FMT procedures in CKD treatment is urgently needed, both clinically and commercially. Clinical trials should evaluate the impact of FMT on hard clinical endpoints in CKD with the assistance of sophisticated artificial intelligence and up-to-date bioinformatics interpretation before successfully moving FMT therapy for CKD into clinical practice.

## Figures and Tables

**Figure 1 nutrients-14-02528-f001:**
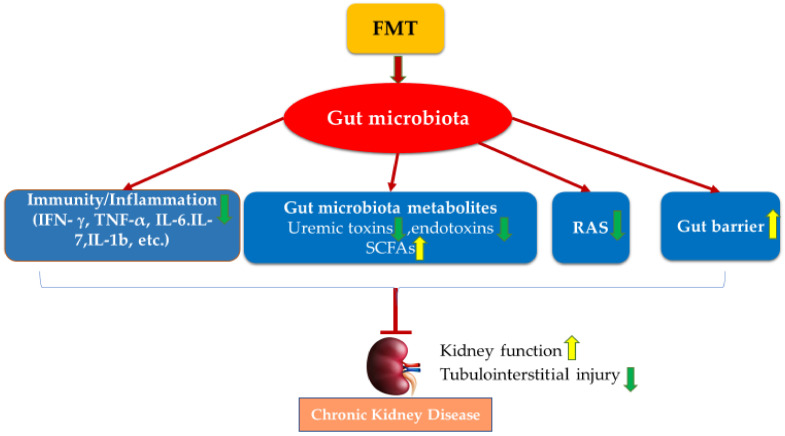
Potential mechanisms of FMT on managing CKD. FMT improves kidney function and protects against kidney injury via the modulation of gut microbiota dysbiosis, mediated by restoring hosts’ immunity, regulating gut microbiota metabolites, inactivating RAS, and improving gut epithelial barrier integrity.

**Table 1 nutrients-14-02528-t001:** Recent pre-clinical and clinical studies on the gut microbial composition in CKD.

Bacteria	Study Type	Disease Type	Alteration Relative to Control	Reference
**Actinobacteria**				
*Bifidobacterium*	Human	CKD	↓	[50]
*Coriobacteriia*	Human	IgAN	↑	[51]
**Bacteroidetes**				
*Prevotellaceae*	Rat	Adenine-induced CKD	↓	[52]
*Alistipes*	Human	IgAN	↓	[53]
**Firmicutes**				
*Faecalibacterium*	Human	DKD	↓	[54,55]
*Eubacterium*	Human	IgAN	↓	[53]
*Ruminnococcaceae*	Rat	Adenine-induced CKD	↓	[52]
*Roseburia*	Human	CKD	↓	[55]
*Clostridia*	Human	IgAN	↓	[53]
*Streptococcus*	Human/Rat	CKD/DKD	↑	[54,55]
**Proteobacteria**				
*Enterobacteriaceae*	Human	CKD	↑	[50]
*E.coli*				
*Proteobacteria*	Human/Rat	DKD	↑	[54,55,56]
*Desulfovibrio*	Human	CKD	↑	[55]
*Escherichia*	Human	IgAN	↑	[51]
*Bacteroidetes*	Human/Mouse	DKD	↓	[54,56]
**Verrucomicrobia**				
Akkermansia	Human	DKD/ESKD	↑	[54,57]
*Verrucomicrobia*	Human/Mouse	DKD	↑	[54,56]

↑: Abundance increased; ↓: Abundance increased.

**Table 2 nutrients-14-02528-t002:** Experimental and clinical studies of FMT in CKD.

Study Type	Disease Models	Administration Method	Outcomes	Ref
Animal study	Adenine-induced mouse model of CKD; 5/6 nephrectomy-induce rat model of CKD	For adenine-induced CKD mice: oral gavage for consecutive 3 days. For 5/6 nephrectomy-induce CKD rats, oral gavage daily for three weeks.	FMT from ESKD increased the production of uremic toxins, aggravated interstitial fibrosis, and oxidative stress in both animal models	[130]
Animal study	STZ-induced DKD mice	150 µL, oral gavage, 3 times on days 1, 2, and 5.	FMT from mice with severe proteinuria led to a higher TMAO and LPS, different microbiota constituents, and more deteriorated kidney damage than those receiving FMT from mice with mild proteinuria.	[131]
Animal study	Db/db mouse model of DKD	Daily oral gavage, once a day for consecutive 7 days.	FMT from resveratrol-treated groups improved kidney functions via anti-inflammation and restored gut microbiota in DKD.	[132]
Animal study	Cyclosporin A-induced mouse model of CKD	Daily oral gavage lasted for 6 weeks from week 7	FMT from *Astragalus membranaceus* (AS)-treated groups attenuated cyclosporin A-induced kidney damage and fatty acid metabolism by improving intestinal barrier, restoring intestinal flora structure, increasing the abundance of probiotics producing butyric acid and lactic acid as well as repairing the disorder of miRNA-mRNA interaction profiles, primarily associated with Butanoate and Tryptophan metabolism.	[133]
Animal study	Adenine-induced murine model of CKD	200 µL daily, once a week for 3 weeks by oral gavage	FMT from healthy mice reduced uremic toxins and improved gut microbiota diversity, but no change in kidney function.	[134]
Animal study	STZ-induced rat model of DKD	Oral gavage once a day for consecutive 3 days.	FMT from healthy control rats effectively alleviated tubulointerstitial injury in diabetic rats by restoring the dysregulated cholesterol homeostasis via activating GPR43.	[135]
Animal study	STZ-induced rat model of DKD	200 μl, oral gavage.	FMT from healthy control rats effectively increased podocyte insulin sensitivity and alleviated glomerular injury in diabetic rats, associated with the downregulation of the GPR43 expression.	[136]
Animal study	BTBR^ob/ob^ mouse model of DKD	300 μl gut microbiota suspension via a rectal route using a polyethylene probe into the intestine.	FMT from BTBR wild-type mice decreased albuminuria and inhibited the overexpression of TNF-α within the ileum and ascending colon in BTBR^ob/ob^ mice.	[137]
Animal study	Cisplatin-induced acute murine kidney injury model; Glycerol-induced murine AKI model; Adeline-induced murine chronic kidney failure model; gentamicin-induced porcine AKI model	For cisplatin-induced acute murine kidney injury model, Glycerol-induced murine AKI model, and gentamicin-induced porcine AKI model, 1 × 10^8^ c.f.u. per mouse, from day 1 to day 10 via intragastric administration (i.g); For Adeline-induced murine chronic kidney failure model, 1 × 10^8^ c.f.u. per mouse, i.g., every two days from day 22 to day 45.	The encapsulated microbial cocktail significantly reduced serum urea and creatinine levels without any adverse effects in AKI and CKD murine and porcine kidney failure models.	[138]
Case study	Membranous nephropathy	Endoscopic administration twice on day 0 and 28.	Membranous nephropathy symptoms were eased, and kidney function was improved.	[139]
Case study	IgA nephropathy	Case 1: 40 times consecutively (200 mL daily, 5 d/week) and then a further 57 times (200 mL daily, 10–15 d/month) over the next 5 months through transendoscopic enteral tubing (TET); Case 2: 60 treatments in 6 months (200 mL daily, 10–15 d/month) via TET and followed up for 6 months.	FMT decreased 24 h urinary protein, increased serum albumin, and restored gut microbiota in both patients	[140]

## Data Availability

Not applicable.

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
