# Peer review of "Faecal Microbiota Transplantation and Chronic Kidney Disease"

_nutrients, 2022, doi:10.3390/nu14122528_

Round 1

Reviewer 1 Report

The subject of the review is very interesting and actual.

In the abstract lines 15&20-21 - please think  about re-writting the sentences, as first Authors declaring the data from pre-clinical and clinical studies are suggesting the promissing microbiota effect on CKD, and then mentioned about limited number of the studies.

line 28 -citation proven the % number is needed.

Please avoid using same sentences in abstract and Introduction part.

In the Introduction part dedicated to description of CKD is very brief, with sudden switch from changes in the kidney to molecular aspects of disease regulation - there is a need of the "bridge" betwwen those two fields.

Please use correct term for: Clostridioides/Clostridium difficile (line 45)

line 69- please give some examples of the diseases

line 72- please add citation dedicated to postbiotics role

lines 78-79; please think about re-writting of the sentence - it is not suggestion of the Authors

line 101 instead of "setting" the reviewer suggests "development" or "onset"

Please check Table 1 format - arrows are shifted

Part 2.2 is quite erratic - would be good to point exact types of kidney diseases in Table 1.

line 187; please explain the abbreviation of  CCL20/CCR6 ligand receptor pair and its role;

line 247 please give examples of food sources

line 339 - please give some more information

Very good Table 2.

line 358 - LPS- please give explanation of used abbreviation

line 369- explanation of latin names needed

line 394 - which bacteria?

not all inflamatory/immune factors are mentioned on Fig.1

line 439 - please add more info about BME

Conclusions are interesting and leading to future needs in the field.

Presenting the results from pre-clinical and clinical studies is a strong point of this review 9animal/human models).

Please underline stronger the connection between diet- gut dysbiosis-CDK.

References list is very actual.

Author Response

Point 1: In the abstract lines 15&20-21 - please think about re-writting the sentences, as first Authors declaring the data from pre-clinical and clinical studies are suggesting the promising microbiota effect on CKD, and then mentioned about limited number of the studies.

Response 1: The sentences have been rewritten as suggested.

Point 2: line 28 -citation proven the % number is needed.

Response 2: The citation has been included in the revised manuscript as suggested.

Point 3: Please avoid using same sentences in abstract and Introduction part.

Response 3: The sentences in the Introduction have been changed to avoid the same sentences in the abstract.

Point 4: In the Introduction part dedicated to description of CKD is very brief, with sudden switch from changes in the kidney to molecular aspects of disease regulation - there is a need of the "bridge" between those two fields.

Response 4: The information has been included in the revised manuscript as suggested.

Point 5: Please use correct term for: Clostridioides/Clostridium difficile (line 45)

Response 5: The term for Clostridium difficile has been corrected in the revised version.

Point 6: line 69- please give some examples of the diseases

Response 6: The information has been included in the revised manuscript as suggested.

Point 7: line 72- please add citation dedicated to postbiotics role

Response 7: The citations have been included in the revised manuscript as suggested.

Point 8: lines 78-79; please think about re-writting of the sentence - it is not suggestion of the Authors

Response 8: The sentences have been rewritten as suggested.

Point 9: line 101 instead of "setting" the reviewer suggests "development" or "onset"

Response 9: The “setting” has been replaced by “onset” as suggested.

Point 10: Please check Table 1 format - arrows are shifted

Response 10: The table has been modified to make sure the arrows are aligned properly.

Point 11: Part 2.2 is quite erratic - would be good to point exact types of kidney diseases in Table 1.

Response 11: The types of kidney diseases have been included in the revised table as suggested. 

Point 12: line 187; please explain the abbreviation of CCL20/CCR6 ligand receptor pair and its role;

Response 12: The information has been included in the revised manuscript as suggested.

Point 13: line 247 please give examples of food sources

Response 13: The information has been included in the revised manuscript as suggested.

Point 14: line 339 - please give some more information

Response 14: The information has been included in the revised manuscript as suggested.

Point 15: line 358 - LPS- please give explanation of used abbreviation:

Response 15: The full name of LPS has been included in the revised manuscript as suggested.

Point 16: line 369- explanation of latin names needed:

Response 16: The information has been included in the revised manuscript as suggested.

Point 17: line 394 - which bacteria?

Response 17: The bacteria information has been included in the revised manuscript as suggested.

Point 18: Not all inflammatory/immune factors are mentioned on Fig.1

Response 18: All the inflammatory/immune factors are included in Fig1.

Point 19: line 439 - please add more info about BME

Response 19: More information about BME has been added in the place where it is fist mentioned.

Reviewer 2 Report

The authors review the potential role for fecal transplantation in the treatment of CKD. They provide extensive background into the potential mechanisms and summarize an enormous body of biochemical, animal, and clinal literature in a logical way that it is easy to follow. This review will serve as a clear roadmap for researchers in this space for years. I commend the authors for their excellent work. Please consider these minor comments when editing the manuscript.

Table 1: I am not sure if it is just my PDF, but the arrows do not line up well with the rows. Please make sure that this is aligned properly in the published table. For the arrow colors, green and blue might be too close to each other, especially since 8% of men have red-green colorblindness.

3.2: paragraph 3: sentence 2: “…supplementation of butyrate mitigates the progression of CKD.” This is not well-established. Butyrate supplementation is not at all a standard of care treatment for CKD. The authors should clarify this.

3.2: paragraph 3: sentence 3: The wording here is difficult to follow. Consider rewording or dividing into two sentences.

The classic teaching for the pathophysiology of CKD is primarily vascular. Tobacco, hypertension and diabetes likely cause irreversible mechanical damage to the renal microvasculature and filtration mechanisms. In the classic CKD patient with tobacco use, hypertension, and diabetes, it seems unlikely that a single fecal transplant will stop the progression of this mechanical damage. I suspect that multiple fecal transplants or probiotic doses would be needed and that they would serve as an adjunct to the management of the underlying causes of CKD to help slow the progression.

Author Response

Point 1: I Table 1: I am not sure if it is just my PDF, but the arrows do not line up well with the rows. Please make sure that this is aligned properly in the published table. For the arrow colours, green and blue might be too close to each other, especially since 8% of men have red-green colour-blindness.

Response 1: The table has been modified to make sure the arrows are aligned properly. The arrow colours have also been changed.

Point 2: 3.2: paragraph 3: sentence 2: “…supplementation of butyrate mitigates the progression of CKD.” This is not well-established. Butyrate supplementation is not at all a standard of care treatment for CKD. The authors should clarify this.

Response 2:  The sentence has been modified in the revised manuscript.

Point 3: 3.2: paragraph 3: sentence 3: The wording here is difficult to follow. Consider rewording or dividing into two sentences.

Response 3:  The sentence has been divided into two sentences as suggested.